# RLE: A Unified Perspective of Data Augmentation for Cross-Spectral Re-identification

**Lei, Tan[1], Yukang Zhang[1], Keke Han[1],**
**Pingyang Dai[1]\*, Yan Zhang[1], Yongjian Wu[2], Rongrong Ji[1]**
[1]Key Laboratory of Multimedia Trusted Perception and Efficient Computing,
Ministry of Education of China, Xiamen University, 361005, P.R. China.
[2]Tencent Youtu Lab, China.
tanlei@stu.xmu.edu.cn, zhangyk@stu.xmu.edu.cn, hankeke303@stu.xmu.edu.cn,
pydai@xmu.edu.cn, bzhy986@xmu.edu.cn, littlekenwu@tencent.com, rrji@xmu.edu.cn

## Abstract

This paper makes a step towards modeling the modality discrepancy in the cross-spectral re-identification task. Based on the Lambertain model, we observe that the non-linear modality discrepancy mainly comes from diverse linear transformations acting on the surface of different materials. From this view, we unify all data augmentation strategies for cross-spectral re-identification by mimicking such local linear transformations and categorizing them into moderate transformation and radical transformation. By extending the observation, we propose a Random Linear Enhancement (RLE) strategy which includes Moderate Random Linear Enhancement (MRLE) and Radical Random Linear Enhancement (RRLE) to push the boundaries of both types of transformation. Moderate Random Linear Enhancement is designed to provide diverse image transformations that satisfy the original linear correlations under constrained conditions, whereas Radical Random Linear Enhancement seeks to generate local linear transformations directly without relying on external information. The experimental results not only demonstrate the superiority and effectiveness of RLE but also confirm its great potential as a general-purpose data augmentation for cross-spectral re-identification. The code is available at https://github.com/stone96123/RLE.

## 1 Introduction

Identity recognition has attracted intensive attention in the last few years due to its wide applications in surveillance systems [1, 2, 3, 4]. Since silicon-based digital cameras are naturally sensitive to near-infrared (NIR), most cameras provide infrared (IR) images instead of visible (VIS) images for better visual quality under poor illumination conditions. In practice, this puts the re-identification (Re-ID) problem in a cross-spectral setting and requires the approaches to properly handle both the intra-class variance and the more significant modality discrepancies between cross-spectral images [5, 6, 7]. Encouraged by the great success of single-modality re-identification, substantial research efforts in cross-spectral re-identification attempt to transform the cross-spectral re-identification challenge into a single-modality learning task. To achieve this goal, previous efforts utilize DNN-based image processing such as Generative Adversarial Networks (GANs) [8], to construct the translation from one spectrum to another. These methods [9, 10] generally provide good visual effects and adjustability. However, the limited visual quality of the generated images and the lack of large-scale databases providing cross-spectral image pairs make GAN training challenging, thus limiting the performance of these methods. Another mainstream strategy focuses on the channel difference between infrared

---

*Corresponding Author.

images [11, 12]. Methods such as grayscale transformation and random channel selection attempt to use image transformation strategies to mimic the transformation between cross-spectral images, thereby pushing the network to adapt to such a transformation. While these methods make sense and decrease the modality discrepancy, lacking the modeling of cross-spectral transformation, they usually tend to pursue the similarity in human visual perception rather than real cross-spectral transformation.

In this paper, we attempt to explore the possibility of modeling the multi-spectral transformation to provide more interpretability, and thus further push the boundary of cross-spectral Re-ID approaches. Based on the Lambertian reflection model [14, 15, 16], we find that the illuminations of the same region in VIS and NIR photos should be able to be described using a simple linear model, as long as the region is composed of one consistent material (details are discussed in Sec. 3). This is illustrated in Figure 1. Here, we use paired VIS-NIR images from the dataset in [13]. For the red and yellow regions in the middle of the image, with a simple linear model, we can accurately predict the pixel values of the NIR image based on the VIS image, as long as the re-

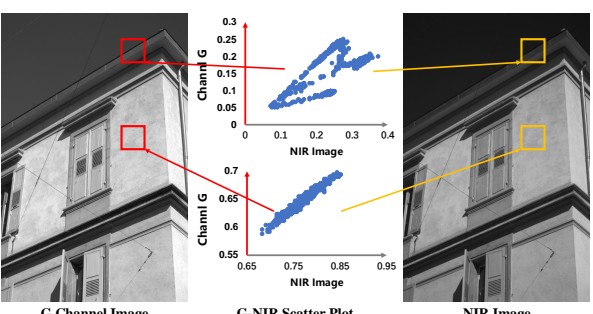

| G Channel Image | G-NIR Scatter Plot | NIR Image |

Figure 1: **Illustration of the cross-spectral transformation. G refers to the green channel of the visible image.** Under the same illumination, the cross-spectral transformation could be described as a linear transformation in a material-similar surface. Still, in the whole image level, the transformation is nonlinear due to the diversity of materials. Since the Re-ID image pairs are not well aligned, we select the cross-spectral image pairs from [13].

gion only has one material. Although the linear transformation exists at the pixel level of cross-spectral image pairs, the material's reflection function determines the linear factor. It means that the linear factor is inconsistent across different surfaces, resulting in an image-level non-linear transformation. In Section 3, we analyze and visualize the result to confirm whether the different linear factors on different surfaces are the main culprit that induces the modality discrepancy in the cross-spectral images. It is interesting to find that the modality discrepancy occurs when using variable linear factors among different patches in the image.

The above observation provides us with a fresh perspective on the cross-spectral Re-ID task. Empirically, adopting observation in image generation seems to be the most intuitive way. As long as we are able to identify regions' materials with their visible or infrared input and calculate the linear coefficients to transform the input image from one spectrum to another, the modality discrepancy would be easy to bridge. Unfortunately, the correlation between visible or infrared input and regional materials is quite limited, which also confines generative strategies in this task to a clear upper bound. Besides exposing the bottleneck of the generative strategy, the observation also provides a unified perspective to rethink the augmentation strategies within this topic. **From this perspective, we discover that data augmentation for cross-spectral re-identification is formed to achieve non-linear transformations with different distinct local linear factors, thus encouraging the network to be robust to such a transformation.** Therefore, under this view, we can easily categorize all the data augmentation strategies designed for cross-spectral re-identification into moderate transformation and radical transformation based on the extent of changes to images. We assign moderate transformation as a strategy that can still keep the original linear correlation after the transformation. Generally, achieving moderate transformation may require precise material labels on each pixel. However, with benefits from the diversity of different channels in visible images, we can obtain a moderate transformation by a linear calculation based on the original image channels. Methods like channel exchange and grayscale transformation are both special cases under moderate transformation. Within the unified formulation of moderate transformation, in this paper, we further provide a more general moderate transformation as Moderate Random Linear Enhancement (MRLE), which aims to use an unfixed mixing of different channels to provide more diverse augmentation results. In contrast to moderate transformations, radical transformations attempt to apply linear transformations to randomly selected local areas. Compared to moderate transformations, which have a limited transformation space and are only effective on multi-channel visible images, radical transformations can produce a more diverse range of results even on single-channel infrared images. However, due to the lack of

constraints, these transformations often introduce additional noise into the original image. Methods such as random erasing [17] and channel random erasing [11] can be considered special cases of radical transformation where the linear factor is set to 0. Similarly, based on the above perspective, we also provide a Radical Random Linear Enhancement strategy, that yields competitive augmentation results by directly applying linear transformations to randomly selected local areas.

In summary, our contributions are threefold:

- As an effort to model the transformation behind the modality discrepancy in the cross-spectral Re-ID task, we discover that the cross-spectral modality discrepancy mainly comes from different local linear transformations caused by the diversity of materials. Based on this observation, we further categorize the cross-spectral data augmentation strategies into moderate and radical transformations under a unified perspective.
- By extending the observation, we propose a Random Linear Enhancement (RLE) strategy, which includes Moderate Random Linear Enhancement (MRLE) and Radical Random Linear Enhancement (RRLE). The RLE effectively takes advantage of the aforementioned unified perspective and embeds it in a controllable linear transformation.
- Extensive experiments on cross-spectral re-identification datasets demonstrate the effectiveness and superior ability of the proposed RLE, which can boost performance under various scenarios.

## 2   Related Works

Cross-spectral re-identification is a challenging task due to the significant modality discrepancy. Two typical frameworks have been proposed to solve such a challenging task. The first one is feature-level learning [18, 19, 20, 21, 22], which aims to bridge the modality gap through well-designed loss functions and end-to-end training. Such a strategy works well in both supervised, semi-supervised, and unsupervised [23] cross-spectral re-identification tasks due to the great power of deep learning. However, these approaches usually do not use any real physics models, making it not uncommon for them to make strange mistakes. To make things worse, due to the high complexity and lack of interpretability, the models are hard to adjust or improve. The other mainstream method to solve cross-spectral re-identification is the image-level strategy, which aims to construct an efficient transformation between different spectrums. Under this condition, the cross-modality discrepancy is considered an individual problem alongside the Re-ID problem. $D^2RL$ [5] makes the first attempt by using variational autoencoders (VAE) for style disentanglement and generates synthesis images from one spectrum to another. AlignGan improves this framework by proposing a unified GAN framework with efficient constraints. Although playing a min-max game between the complex generator and discriminator offers visually impressive results, the generated images are still far from photorealistic, and in turn, limit the final performance. Therefore, these methods were subsequently superseded by lighter-weight modality generate strategies. This improvement suggests that cross-spectral transformations may not be as complicated as previously envisaged. X-modality [24] designs a lightweight network to learn an intermediate mediator from visible images, while MMN [25] improves it by extending an infrared side. Recently, CAJ [11] and CAJ+ [12] directly removed the extra generator and utilized several types of grayscale images as an assistant for training which also achieves satisfying performance. Although recent methods have made some progress in this topic, due to the lack of analysis and modeling for cross-spectral transformation, the methods tend to pursue the similarity of transformation in human visual perception rather than real cross-spectral transformation.

## 3   Reflection Prior for Cross-Spectral Images

VIS-NIR matching is a longstanding computer vision problem that has been explored for decades [26, 27]. One of the main challenges is to formulate and thus alleviate the modality discrepancy. Using the Lambertian model to analyze the digital image from multi-sensor cameras is widely applied in some pioneer works [15, 28]. With a light source emitting photons across different wavelengths $\lambda$, the response of each pixel $(x, y)$ in the camera sensors can be formulated as:

$$\rho_j(x,y) = \sigma(x,y) \int_{\lambda_j} E_j(\lambda, x, y) S(\lambda, x, y) Q_j(\lambda) d\lambda, \tag{1}$$

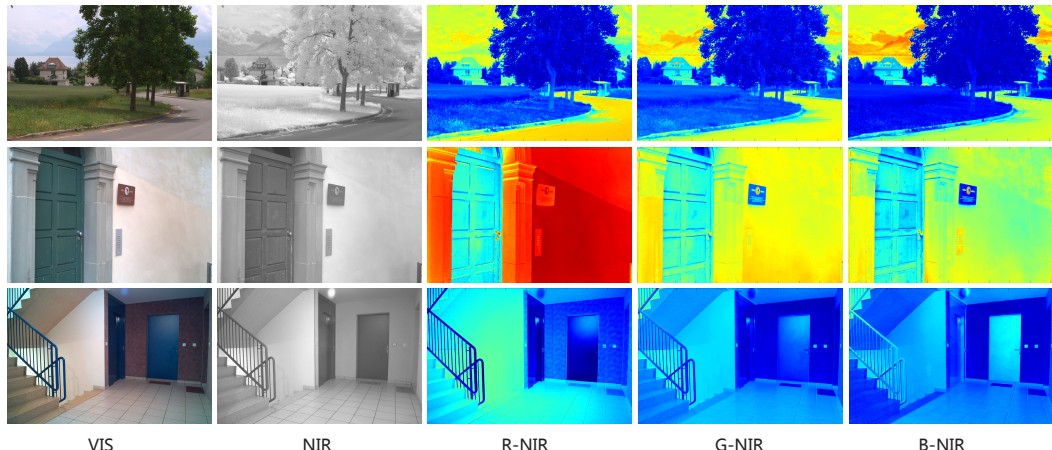

| VIS | NIR | R-NIR | G-NIR | B-NIR |

Figure 2: **Example images from the VIS-NIR scene dataset [13].** After we divide the visible image into the red, green, and blue channels and form chromaticity band ratios from these three spectra and the NIR image, it is clear that the ratio for pixels from the surface with high material-similarity is nearly constant.

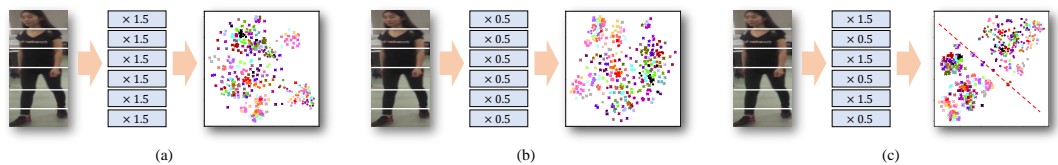

| (a) | (b) | (c) |

Figure 3: **A example about how modality discrepancy occurs.** Feature space visualization of 100 randomly selected images with (dot) and without (fork) the local linear transformation on the original image. **(a)**∼**(b)**: The same linear factor takes effect on the whole image bringing limited modality discrepancy. **(c)**: Variable linear factors take effect on different parts showing a huge modality discrepancy. The 'cross' and 'dot' marks indicate the samples from the original one and the generated one respectively.

where $\lambda$ is the wavelength, as well as $E(\lambda)$ and $S(\lambda)$ denote the spectral power distribution (SPD) of incident light and surface spectral reflectance. $Q(\lambda)$ is the spectral sensitivity of the camera sensor. $j = \{R, G, B, N\}$ indicates the channel (spectrum). $\sigma(x, y)$ is the Lambertian reflection term which is a constant factor and can be calculated by the dot product of the surface normal with the illumination direction.

Following Eq. (1), we leverage a mild assumption to derive a representation between the SPD of the light source and incident light. Generally, we could describe the SPD of the light source by a relative spectral power distribution $F(\lambda, x, y)$ together with a variable $\omega$ that reflects the illumination intensity. We assume that the SPD of incident light in the whole image keeps the same relative spectral power distribution as the light source. Then we could formulate the $E(\lambda, x, y)$ as:

$$E_j(\lambda, x, y) = \beta_j(x, y)\omega_j F_j(\lambda), \tag{2}$$

where $\beta$ is a parameter to reflect the ratio of intensity between the incident light and the light source. Then from Eq. (1) and Eq. (2), if we now consider the images under different spectra, such as G channel images and NIR images, it is clear that the transformation of G-NIR could be described as:

$$\frac{\rho_N(x, y)}{\rho_G(x, y)} = \frac{\omega_N \beta_N(x, y) \int_{\lambda_N} F_N(\lambda) S(\lambda, x, y) Q_N(\lambda) d\lambda}{\omega_G \beta_G(x, y) \int_{\lambda_G} F_G(\lambda) S(\lambda, x, y) Q_G(\lambda) d\lambda}. \tag{3}$$

Under an ideal condition, $\beta$ is supposed to be a high-order term determined by the distance between the light source and the surface [29]. Now, we could utilize this approximation and regard $\beta$ as a constant under the same light source to get a simplified expression:

$$\frac{\rho_N(x, y)}{\rho_G(x, y)} = \frac{\omega_N M(x, y, N)}{\omega_G M(x, y, G)}. \tag{4}$$

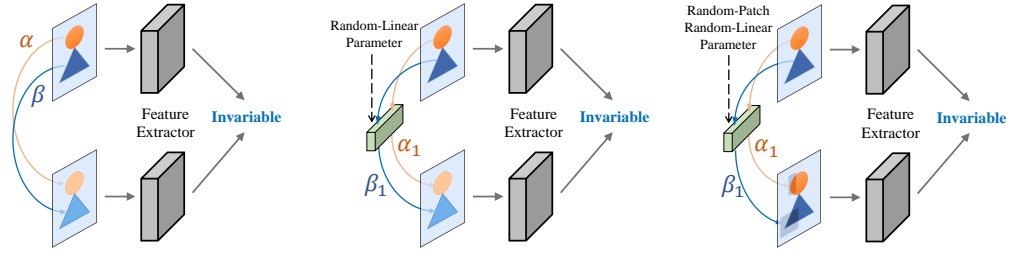

(a) Definite linear transformation on Definite image patch. (b) Random linear transformation on Definite image patch. (c) Random linear transformation on Random image patch.

Figure 4: **The motivation of RLE.** Herein, we construct an ideal person with only two different surfaces and ignore the background. **(a)**: As demonstrated above, to obtain a spectral-invariant feature representation, the network should be robust to such a transformation that takes effect upon definite surfaces by definite linear factors. **(b)**: An ideal data augmentation strategy that takes effect upon definite surfaces by random linear factors. However, this method needs a hard-achieved extra material-aware network for segmentation. **(c)**: The idea of RRLE. By taking effect upon random surfaces by random linear factors, the RRLE encourages the network to be robust to a linear transformation anywhere in the image. Under this condition, the cross-spectral transformation can be considered as an easy state of RRLE space.

Since $F(\lambda)$, $Q(\lambda)$, and $S(\lambda)$ are three inner functions depending on the SPD of the light source, the sensitivity of the camera sensor, and the reflection function of the surface material, we replace the Riemann integral with a function $M(x, y, j)$. In addition, $\frac{\omega_N}{\omega_G}$ could be considered as a constant factor in two determined spectra. From this representation, one could observe that the cross-spectral transformation is a linear transformation in those regions of the same material and under the same illumination condition, as shown in Figure 1. If we further extend it to the entire image, the factor is only influenced by $S(\lambda, x, y)$ which is determined by the material.

To verify whether the above equation could be used in various real-world scenarios, we used the paired VIS-NIR scene image dataset introduced by [13]. In Figure 2, we form chromaticity band ratios between three VIS spectra and NIR spectrum at each pixel and use the color to reflect the ratio. We discovered that the ratio is nearly constant within a region with a consistent material, which holds across R, G, B, and NIR spectra.

After observing the above linear transformation, we further explore whether the variable linear factor in different surfaces is the main culprit that induced the modality gap in such an application. Due to the lack of material labels that are available to guide sample generation, we uniformly segmented 100 randomly selected images into six parts from the top to the bottom and multiplied each part by a linear factor. Then, we send the new images and original images into an ImageNet [30] pre-trained Resnet-50 [26]. Although not so well-aligned, benefiting from the body structure prior from head to toe, we still find that the modality discrepancy occurs when suffering from variable linear factors.

## 4  Random Linear Enhancement

From the above observation, we can unify all data augmentation strategies for cross-spectral re-identification as mimicking such a local linear transformation, thereby encouraging the network to be robust to transformation, as shown in Figure 4 (a). Based on this perspective, by considering the influence on the original image, the data augmentation strategies can easily be categorized into two types: moderate transformation and radical transformation. By extending the observation, this paper pushes the boundary of both types by proposing a Random Linear Enhancement strategy (RLE).

### 4.1  Moderate Random Linear Enhancement

As mentioned before, we assign moderate transformation as a strategy that maintains the original linear correlation after the transformation. Benefiting from the difference between different channels (R, G, B) in visible images, we can obtain a moderate transformation without precise material labels.

From this view, the processing of moderate transformation can be unified as:

$$I_{mt} = \lambda_r I_r + \lambda_g I_g + \lambda_b I_b,$$
$$s.t. \quad \lambda_r + \lambda_g + \lambda_b = 1,$$

(5)

where the $I_{mt}$ indicates the transformed image, as well as $I_r$, $I_g$, and $I_b$ refer to the red, green, and blue channels of the visible image, respectively. Here, $\lambda_r$, $\lambda_g$, and $\lambda_b$ are hyper-parameters to control the mixing percentage. It is evident from Eq. (5) that random channel selection corresponds to the specific cases where the parameters $\lambda_r$, $\lambda_g$, and $\lambda_b$ are specified as $[1, 0, 0]$, $[0, 1, 0]$, or $[0, 0, 1]$. Also, the grayscale transformation is the specific cases where the parameters $\lambda_r$, $\lambda_g$, and $\lambda_b$ are specified as $[0.299, 0.587, 0.114]$.

It is apparent that previous strategies exhibited significant limitations in their parameter settings, leading to highly restricted augmentation results. Therefore, we attempt to relax the settings of $\lambda_r$, $\lambda_g$, and $\lambda_b$ and propose a Moderate Random Linear Enhancement (MRLE). Generally, sampling from a uniform distribution to determine the values of three hyper-parameters is identified as the simplest and most efficient approach. However, while a uniform distribution uniformly covers the entire feasible transformation domain, in practice, those samples at the boundaries always contribute more to learning decision boundaries. Consequently, we employ a U-shaped beta distribution instead of a uniform distribution for hyper-parameter sampling. This not only maintains the feasible transformation domain but also enhances the sampling probability of boundary samples. In general, the formulation of MRLE can be given as follows:

$$I_{mt} = \lambda_r I_r + \lambda_g I_g + \lambda_b I_b,$$
$$with \quad \lambda_r, \lambda_g, \lambda_b \sim Beta(\beta_m, \beta_m),$$
$$s.t. \quad \lambda_r + \lambda_g + \lambda_b = 1.$$

(6)

Herein, $\beta_m$ is the hyper-parameter to control the shape of the beta distribution.

## 4.2 Radical Random Linear Enhancement

Although MRLE can provide diverse transformation results that obey the original linear correlation in the image, it can only take effect on the multi-channel visible image and shows quite limited transformation space. The ideal data augmentation appears to be using random linear factors on different surfaces as shown in Figure 4 (b). However, this approach heavily relies on pixel-level material labels, which are hard to obtain. Therefore, achieving such a local linear transformation without adequate material labels may inevitably involve some risk-taking. To achieve this goal, as shown in Figure 4 (c), we propose the Radical Random Linear Enhancement (RRLE) that randomly selects several image patches and multiplies them with a variable linear parameter to directly mimic the local linear transformation. Under the RRLE, the cross-spectral transformation could be considered as a sub-state of the whole state space.

Concretely, for an input image $I$, the RRLE randomly selects a rectangle region $I_{select}$ following the same setting of random erasing [17] and multiplies it with a linear factor $\alpha$. In case it may exceed the upper bound, we calculate the maximum feasible linear factor in $I_{rle}$ as $\alpha_{max}$. The linear factor $\alpha$ is calculated by multiplying $\alpha_{max}$ and a random factor $f_g$ between 0 to 1. Typically, small linear factors may not be sufficient to effectively provide enough variation in the original image[2]. Therefore, a U-shaped Beta distribution is utilized for $f_g$ to obtain and provide high-quality training samples. In general, the formulation of RRLE can be given as follows:

$$I_{rt} = \alpha I_{select},$$
$$with \quad \alpha = \alpha_{max} f_g,$$
$$and \quad f_g \sim Beta(\beta_r, \beta_r).$$

(7)

Herein, $I_{rt}$ indicates the transformed selected region. $\beta_r$ refers to the hyper-parameter to control the shape of the beta distribution.

Furthermore, following the setting of random erasing, we set $s_{min}$ and $s_{max}$ to control the area of the selected region, while setting $r_{min}$ and $r_{max}$ to adjust the aspect ratio. Unlike most data augmentation strategies that take effect on the image only once, the RRLE will be repeated several

---

[2]Visualization results are provided in the appendix.

times to obtain a higher modality discrepancy. Since the repeat will bring extra noise, we set a memory matrix $M$ to store the cumulative changes at each pixel. We set a $t_{min}$ to terminate the RRLE when $min(M) < t_{min}$. To better explain the processing, we provide detailed procedure of RRLE in the appendix.

# 5 Experiments

## 5.1 Datasets and Implementation details

We conduct experiments on two publicly available visible-infrared person re-identification datasets SYSU-MM01 [31] and RegDB [32].

**SYSU-MM01** is a large-scale dataset captured by four visible cameras and two infrared cameras in both indoor and outdoor environments. The training set contains 395 identities with $22,258$ visible images and $11,909$ infrared images, while the testing set includes 96 identities with $3,803$ infrared images as the query. This dataset contains two different search modes, the all-search mode and the indoor-search mode. In the all-search mode, the gallery images are from all the visible cameras. For the indoor-search mode, the source of the gallery set excludes two outdoor cameras.

**RegDB** dataset is collected by two aligned cameras, one for visible and the other for far-infrared (thermal). It contains $412$ identities, each with $10$ visible images and $10$ infrared images. Following the evaluation protocol of previous works [33, 34], we choose half of the identities at random for training and the other half for testing. The results are the average of 10 repeating.

We follow the evaluation settings in existing VI-ReID methods [11, 1, 12] and adopt the Cumulative Matching Characteristic (CMC), mean Average Precision ($m$AP) and mean Inverse Negative Penalty (mINP) as evaluation metrics.

## 5.2 Implementation details

We use Pytorch to implement our method and finish all the experiments on a single RTX 3090 GPU. The mini-batch size is set to 48. For each mini-batch, we randomly select 4 identities, each with 6 visible images and 6 infrared images. We select the ResNet-50-based PCB [35] with the global branch as the baseline, which is a widely used fine-grid part feature learning framework in both Re-ID and visible-infrared Re-ID [25, 18]. We also divide the first convolutional layer to tackle the two modalities' input as usual [36, 37]. We resize all of the images to $384 \times 192$ and use random flipping as basic data augmentation. The initial learning rate is set to 0.1, and decayed by 0.1 and 0.01 at 20, and 50 epochs. Following previous works[38, 11, 39], we apply a warm-up strategy in the first 10 epochs. To better verify the ability of the proposed data augmentation strategy, we just use the basic softmax cross-entropy loss and triplet loss during the training without adding any extra constraints to solve the modality discrepancy.

## 5.3 Ablation Study

In this section, we conduct empirical experiments to show the performance under different data augmentation strategies. Since we have categorized all data augmentation strategies for cross-spectrum re-identification into moderate transformations and radical transformations, we have conducted relevant discussions on these two aspects and mixed transformations.

**Moderate transformation.** Here, we evaluate the influence of the performance with different moderate transformation strategies and show the quantitative results in Table 1. In particular, we compare the proposed MRLE with the widely used grayscale transformation ('Gray') and random channel selection ('RC'). In previous works [11, 12], the 'Gray' and 'RC' are usually used together to obtain more diverse results. Therefore, we also give the result under both 'Gray' and 'RC'.

Compared to the baseline, every moderate transformation yielded positive gains showing the effectiveness of moderate transformation. However, although methods such as 'RC' and 'Gray' do simulate cross-spectrum transformations to a degree, their restricted transformation spaces result in smaller performance enhancements compared to MRLE. By fully exploring the feasible transformation space, MRLE managed to surpass the limits of earlier moderate transformation strategies, achieving a significant increase in performance in all metrics.

Table 1: Ablation study of different data augmentation strategies on the cross-spectral re-identification task. 'Gray' denotes the grayscale transformation, 'RC' refers to the random channel selection. 'MRLE' indicates the moderate random linear enhancement. 'RE' refers to the random erasing, and 'RRLE' means the radical random linear enhancement.

| Setting | All Search | | | | | | Indoor Search | | | | | |
|---|---|---|---|---|---|---|---|---|---|---|---|---|
| | R-1 | R-5 | R-10 | R-20 | mAP | mINP | R-1 | R-5 | R-10 | R-20 | mAP | mINP |
| **Moderate Transformation** | | | | | | | | | | | | |
| Baseline | 64.5 | 88.1 | 94.2 | 98.1 | 62.9 | 50.4 | 70.0 | 91.7 | 96.6 | 99.1 | 75.1 | 71.1 |
| Baseline+Gray | 66.7 | 89.4 | 95.2 | 98.7 | 64.2 | 50.8 | 72.0 | 93.9 | 97.8 | 99.6 | 77.1 | 73.0 |
| Baseline+RC | 68.3 | 90.6 | 95.6 | 98.6 | 65.3 | 51.9 | 72.7 | 93.9 | 97.6 | 99.7 | 77.7 | 73.7 |
| Baseline+RC+Gray | 68.6 | 90.7 | 96.0 | 98.8 | 64.9 | 52.3 | 74.3 | 94.1 | 98.1 | 99.6 | 78.8 | 75.1 |
| Baseline+MRLE | **70.2** | **91.6** | **96.5** | **99.0** | **67.0** | **53.5** | **75.5** | **95.2** | **98.2** | **99.7** | **79.7** | **75.9** |
| **Radical Transformation** | | | | | | | | | | | | |
| Baseline+RE | 71.0 | 91.4 | 96.3 | 99.1 | 69.5 | 57.4 | 78.5 | 95.8 | 98.7 | 99.8 | 82.1 | 78.4 |
| Baseline+RRLE | 72.0 | 92.4 | 97.2 | 99.4 | 69.1 | 56.3 | 77.0 | 96.4 | **99.1** | **99.9** | 81.4 | 77.7 |
| Baseline+RE+RRLE | **74.2** | **93.0** | **97.4** | **99.5** | **71.8** | **60.4** | **81.7** | **96.7** | **99.1** | **99.9** | **84.5** | **81.2** |
| **Mixed Transformation** | | | | | | | | | | | | |
| Baseline+CAJ [11] | 73.5 | 92.9 | 97.4 | 99.4 | 69.4 | 55.4 | 80.7 | 96.1 | 98.6 | 99.8 | 83.5 | 79.8 |
| Baseline+RLE+RE | **75.4** | **93.5** | **97.7** | **99.6** | **72.4** | **60.9** | **84.7** | **97.9** | **99.3** | **99.9** | **87.0** | **83.7** |

**Radical transformation.** Besides the moderate transformation, we also provide a detailed empirical study of the radical transformation including random erasing ('RE') and RRLE in Table 1. We can observe that due to a more flexible transformation space, radical transformations reach an even better performance than the best moderate transformation MRLE.

Although 'RE' can be considered a special case of RRLE with a linear factor of 0, RRLE encourages images to undergo more transformations while preventing the loss of information. Therefore, RRLE and RE tend towards different valuation perspectives and can be used together. As shown in Table 1, the peak performance under radical transformation is reached when combining both 'RE' and RRLE.

**Mixed transformation.** Given that moderate and radical transformations do not conflict formally, they can be combined during the training. Accordingly, we present the performances under a mixed transformation in Table 1. The results indicate that moderate transformations and radical transformations can be used simultaneously and lead to significant performance improvements. Also, for better comparison, we add the recently proposed mixed transformation CAJ which combines the grayscale transformation and random erasing strategy. It is shown that the proposed RLE also works better than the CAJ in cross-spectral re-identification task.

## 5.4 Comparison with State-of-the-Arts

In Table 2, we combine the 'RLE+RE' with a basic framework and evaluate it against the previously reported state-of-the-art methods on the SYSU-MM01 and RegDB. Compared to previous works, it is worth noticing that the basic network doesn't have any extra modules or constraints to cope with the modality discrepancy in cross-spectral re-id. Just combining the basic network with the 'RLE+RE' can achieve comparable performance with state-of-the-art methods, which indicates the great adaptability of RLE in the cross-spectral re-id task.

## 5.5 Discussion

**Hyper-parameter settings of RLE.** RLE contains several hyper-parameters to ensure its effectiveness, such as the $\beta_m$ in Eq. (6), as well as the $\beta_r$ and $t_{min}$ in Eq. (7). Therefore, this part evaluates the performance under different hyper-parameter settings and shows the results in Table 3.

Compared to a uniform distribution, sampling from a U-shape beta distribution performs better in MRLE. The optimal reach when $\beta_m$ is set to 0.3. For RRLE, as a radical transformation, the boundary is more sensitive. Over-transformation may easily destroy the original image. In this framework, peak performance is achieved when $\beta_r = 0.4$ and $t_{min} = 0.1$.

**Applicability of RLE for other methods.** Besides the basic framework employed above, we also explored the integration of the proposed RLE strategy with the current state-of-the-art method to evaluate its extensive adaptability in cross-spectral re-identification tasks.

Table 2: Comparisons between the proposed method and some state-of-the-art methods on the SYSU-MM01 and RegDB datasets.

| Methods | SYSU-MM01 | | | | | | RegDB | | | | | |
| | All Search | | | Indoor Search | | | VIS to IR | | | IR to VIS | | |
| | R-1 | R-10 | mAP | R-1 | R-10 | mAP | R-1 | R-10 | mAP | R-1 | R-10 | mAP |
|---|---|---|---|---|---|---|---|---|---|---|---|---|
| BDTR[40] | 17.0 | 55.4 | 19.7 | - | - | - | 33.6 | 58.6 | 32.8 | 32.9 | 58.5 | 32.0 |
| D$^2$RL[5] | 28.9 | 70.6 | 29.2 | - | - | - | 43.4 | 66.1 | 44.1 | - | - | - |
| Hi-CMD[41] | 34.9 | 77.6 | 35.9 | - | - | - | 70.9 | 86.4 | 66.0 | - | - | - |
| AlignGAN[10] | 42.4 | 85.0 | 40.7 | 45.9 | 87.6 | 54.3 | 57.9 | - | 53.6 | 56.3 | - | 53.4 |
| DDAG[36] | 54.8 | 90.4 | 53.0 | 61.0 | 94.1 | 68.0 | 69.3 | 86.2 | 63.5 | 68.1 | 85.2 | 61.8 |
| LbA[42] | 55.4 | - | 54.1 | 58.5 | - | 66.3 | 74.2 | - | 67.6 | 67.5 | - | 72.4 |
| NFS[43] | 56.9 | 91.3 | 55.5 | 62.8 | 96.5 | 69.8 | 80.5 | 91.6 | 72.1 | 78.0 | 90.5 | 69.8 |
| CM-NAS[44] | 60.8 | 92.1 | 58.9 | 68.0 | 94.8 | 52.4 | 82.8 | 95.1 | 79.3 | 81.7 | 94.1 | 77.6 |
| MCLNet[37] | 65.4 | 93.3 | 62.0 | 72.6 | 97.0 | 76.6 | 80.3 | 92.7 | 73.1 | 75.9 | 90.9 | 69.5 |
| FMCNet[45] | 66.3 | - | 62.5 | 68.2 | - | 74.1 | 89.1 | - | 84.4 | 88.4 | - | 83.9 |
| SMCL[46] | 67.4 | 92.9 | 61.8 | 68.8 | 96.6 | 75.6 | 83.9 | - | 79.8 | 83.1 | - | 78.6 |
| DART[20] | 68.7 | 96.4 | 66.3 | 72.5 | 97.8 | 78.2 | 83.6 | - | 75.7 | 82.0 | - | 73.8 |
| CAJ[11] | 69.9 | 95.7 | 66.9 | 76.3 | 97.9 | 80.4 | 85.0 | 95.5 | 79.1 | 84.8 | 95.3 | 77.8 |
| MPANet[47] | 70.6 | 96.2 | 68.2 | 76.7 | 98.2 | 81.0 | 82.8 | - | 80.7 | 83.7 | - | 80.9 |
| MMN [25] | 70.6 | 96.2 | 66.9 | 76.2 | 97.2 | 79.6 | 91.6 | 97.7 | 84.1 | 87.5 | 96.0 | 80.5 |
| MAUM [48] | 71.7 | - | 68.8 | 77.0 | - | 81.9 | 87.9 | - | - | 87.0 | - | 84.3 |
| CAJ+ [12] | 71.5 | 96.2 | 68.2 | 78.4 | 98.4 | 82.0 | 85.7 | 95.5 | 79.7 | 84.9 | 95.9 | 78.6 |
| DEEN [39] | 74.7 | 97.6 | 71.8 | 80.3 | 99.0 | 83.3 | 91.1 | 97.8 | 85.1 | 89.5 | 96.8 | 83.4 |
| **Ours** | **75.4** | **97.7** | **72.4** | **84.7** | **99.3** | **87.0** | **92.8** | **97.9** | **88.6** | **91.0** | **97.5** | **86.6** |

Table 3: Hyper-parameter settings of RLE. The optimal performance reaches when $\beta_m$, $\beta_r$, and $t_{min}$ is set to $0.3$, $0.4$, and $0.1$ respectively.

(a) **Performance under different** $\beta_m$. Compared to the uniform distribution, a U-shaped beta distribution works better in MRLE.

| $\beta_m$ | R-1 | mAP | mINP |
|---|---|---|---|
| 1.0 | 67.9 | 65.3 | 52.2 |
| 0.5 | 67.8 | 65.1 | 51.8 |
| 0.4 | 68.3 | 65.6 | 52.3 |
| 0.3 | **70.2** | **67.0** | **53.5** |
| 0.2 | 67.9 | 66.0 | 52.6 |

(b) **Performance under different** $\beta_r$. Compared to the uniform distribution, a U-shaped beta distribution works better in MRLE.

| $\beta_r$ | R-1 | mAP | mINP |
|---|---|---|---|
| 0.5 | 72.5 | 70.7 | 59.3 |
| 0.4 | **74.2** | **71.8** | **60.4** |
| 0.3 | 73.2 | 71.3 | 60.2 |
| 0.2 | 73.7 | 71.7 | 60.2 |

(c) **Performance under different** $t_{min}$. Using too small $t_{min}$ will introduce excessive noise and lead to performance degradation.

| $t_{min}$ | R-1 | mAP | mINP |
|---|---|---|---|
| 0.3 | 72.9 | 70.9 | 58.2 |
| 0.2 | 73.8 | 71.3 | 59.0 |
| 0.1 | **74.2** | **71.8** | **60.4** |
| 0.01 | 73.8 | 71.7 | 59.8 |
| 0.001 | 73.6 | 71.3 | 59.5 |

Herein, we add the RLE in the open-sourced method DEEN [39] and show the result in Table 4. Specifically, since the DEEN already contains the random grayscale and random erasing for data augmentation, we remove the random grayscale and add the RLE. Although the DEEN already contains strong augmentations, adding RLE can also bring a performance gain. Beyond the CNN models, we also investigated whether RLE could be applied to a ViT-based structure.

Table 4: Applicability of our opposed RLE to other methods on the SYSU-MM01 dataset.

| Setting | All Search | | Indoor Search | |
| | R-1 | mAP | R-1 | mAP |
|---|---|---|---|---|
| DEEN [39] | 74.7 | 71.8 | 80.3 | 83.3 |
| +Ours | **76.2** (+1.5) | **73.0** (+1.2) | **83.2** (+2.9) | **85.3** (+2.0) |
| ViT-B [49] | 66.0 | 63.1 | 69.9 | 75.1 |
| +Ours | **70.2** (+4.2) | **66.7** (+3.6) | **71.9** (+2.0) | **76.4** (+1.3) |

Since there is no open-source ViT model for cross-spectral re-id, we use the vanilla ViT-B with random erasing augmentation as the basic framework in this part. From Table 4, we can observe that RLE can still work well in a ViT structure. To ensure the generalization ability of RLE, when applying it to other methods, we keep the same hyperparameters setting of RLE with the previous experiments. Therefore, better performance may be achieved on specific methods by fine-tuning the hyperparameters.

**Visualization results of RLE.** To gain a deeper understanding of RLE processing, we visualize the RLE augmented images from both the visible and infrared sides in Figure 5. It can be seen that MRLE provides an efficient way to provide diverse transformations from multi-spectral images to

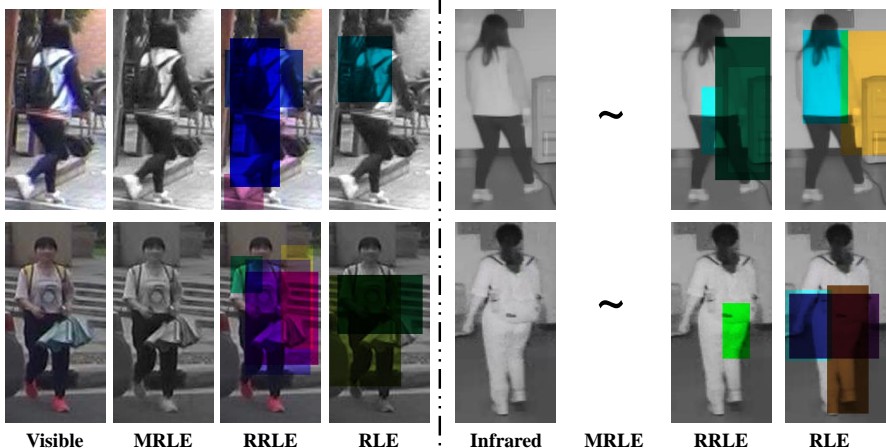

| Visible | MRLE | RRLE | RLE | Infrared | MRLE | RRLE | RLE |

Figure 5: Visualization results of RLE. Since the MRLE can not take effect on the infrared images, we use '~' instead. Meanwhile, Both MRLE and RRLE are used with a certain probability. Therefore, all of the augmentation images above are potential results.

single-spectral images, while the RRLE gets rid of the dependence on the multiple spectral images and makes such a linear transformation directly on the local part. In general, adding such a random linear transformation in the local area of the images largely breaks the color information of the image while preserving the semantic.

## 6 Limitations and Broader Impact

Based on the specific observation in the cross-spectral re-identification, the proposed RLE may not be as general as a data augmentation strategy like random flipping. Whether breaking the modality-similarity between the image pairs could make sense in other computer vision tasks still needs to be evaluated. Meanwhile, under extremely bad weather, such as heavy rain, fog, or limited illumination, the Lambertain model may not work well. So, whether RLE can still perform well in these complex weather is ambiguous. On the other hand, the RegDB and SYSU-MM01 datasets are limited in scale and environment. Although the proposed RLE shows a strong ability to boost the methods in both two datasets, the performance of RLE in an open-world scenario has not yet been verified. Nevertheless, we still believe that the proposed RLE can boost the research of image generation and data augmentation on more general cross-spectral scenarios.

## 7 Conclusion

This paper provides a unified perspective on data augmentation strategies for cross-spectral re-identification. We observe the non-linear modality discrepancy mainly comes from the diverse linear transformation taking effect on different material surfaces; all data augmentation strategies for cross-spectral re-identification aim to simulate this kind of transformation. By extending the observation, we introduce a more general augmentation Random Linear Enhancement (RLE), further pushing the boundary of moderate transformation by Moderate Random Linear Enhancement (MRLE) and radical transformation by Radical Random Linear Enhancement (RRLE). Experimental results show that RLE is effective and applicable in cross-spectral re-identification tasks.

**Acknowledgements.** This work was supported by the National Key R&D Program of China (No.2022ZD0118202), the National Science Fund for Distinguished Young Scholars (No.62025603), the National Natural Science Foundation of China (No.U21B2037, No. U22B2051, No. 62176222, No. 62176223,No. 62176226, No. 62072386, No. 62072387, No. 62072389, No. 62002305, and No. 62272401), and the Natural Science Foundation of Fujian Province of China (No.2021J01002, No.2022J06001).

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

# A Appendix / supplemental material

## A.1 Additional Explanation of Radical Random Linear Enhancement

---

**Algorithm 1:** Radical Random Linear Enhancement

---

**Input:** $I$: Input image; $C$, $H$ and $W$: Image channel and size; $p$: Probability of the LTG; $s_{min}$
      and $s_{max}$: Area of the selected region; $r_{min}$ and $r_{max}$: Aspect of the selected region;
      $t_{min}$: Terminate the LTG;
**Output:** Enhanced image $I^*$
**Initialization:** $p_1 \leftarrow \text{Rand}(0,1)$;
**if** $p_1 \geq p$ **then**
   |  $I^* \leftarrow I$;
   |  return $I^*$;
**else**
   |  $M = Ones(C, H, W)$;
   |  **while** $True$ **do**
   |    |  $S_r \leftarrow \text{Rand}(s_{min}, s_{max}) \times W \times H$;
   |    |  $r_r \leftarrow \text{Rand}(r_{min}, r_{max})$;
   |    |  $H_r \leftarrow \sqrt{S_r \times r_r}$; $W_r \leftarrow \frac{S_r}{r_r}$;
   |    |  $x_r \leftarrow \text{Rand}(0, W)$; $y_r \leftarrow \text{Rand}(0, H)$;
   |    |  **if** $x_r + W_r \leq W$ *and* $y_r + H_r \leq H$ **then**
   |    |    |  $I_{select} \leftarrow (C, x_r, y_r, x_r + W_r, y_r + H_r)$;
   |    |    |  $M_{select} \leftarrow (C, x_r, y_r, x_r + W_r, y_r + H_r)$;
   |    |    |  **for** $i \leftarrow 0$ **to** $C$ **do**
   |    |    |    |  $\alpha_{max} \leftarrow \frac{1}{max(I_{select})}$;
   |    |    |    |  $\alpha \leftarrow \alpha_{max} \times f_g$;
   |    |    |    |  $I(I_c) \leftarrow \alpha \times I_c$;
   |    |    |    |  $M(M_c) \leftarrow \alpha \times M_c$;
   |    |    |  **end**
   |    |  **end**
   |    |  **if** $min(M) \leq t_{min}$ **then**
   |    |    |  $I^* \leftarrow I$;
   |    |    |  **Return** $I^*$
   |    |  **end**
   |  **end**
**end**

---

In Radical Random Linear Enhancement (RRLE), we use a U-shape beta distribution instead of the uniform distribution to generate the linear factor. Here, we show an example of modality discrepancy under different linear factors. Following the above setting, we uniformly segmented 100 randomly selected images into six parts from the top to the bottom and multiplied each part by a linear factor. Then, we send the new images and original images into an ImageNet [30] pre-trained Resnet-50 [26] and visualization of the feature space. As shown in Figure. 6, when using a small linear factor may not be enough to bring a significant modality gap. Thus, we use a U-shaped beta distribution to drive more dramatic changes in the linear factors.

Meanwhile, in this section, we provide a detailed presentation of the RRLE, including a detailed procedure of the RRLE in Alg. 1.

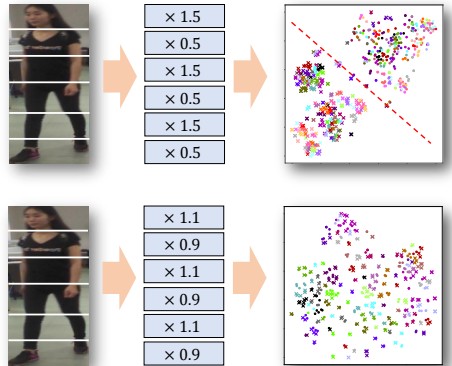

Figure 6: **A example of modality discrepancy.** The dot and forks refer to the sample with and without linear transformation. Clearly, Small linear factors may not be so efficient in generating images with a significant modality gap in the training stage.

