# OpenReview forum: "RLE: A Unified Perspective of Data Augmentation for Cross-Spectral Re-Identification"
_NeurIPS.cc/2024/Conference — NeurIPS 2024 poster_

### Official Review · Reviewer_DZyN · 2024-06-28

**Soundness:** 3
**Presentation:** 3
**Contribution:** 3
**Rating:** 8
**Confidence:** 5

**Summary:**

This paper provides a unified perspective to consider the data augmentation strategies in cross-spectral re-identification. Based on the Lambertain model, the author finds that the cross-spectral discrepancy is induced by multiple local linear transformations. Furthermore, the authors propose a robust linear enhancement (RLE) to imitate such a transformation moderately and radically. Visualization and experimental results in related datasets show the proposed RLE's effectiveness in cross-spectral circumstances.

**Strengths:**

1. The motivation of this paper seems to be valid and intriguing, and the proposed unified perspective makes sense in this topic.
2. The paper is easy to follow and well-written.
3. The proposed method shows superior performance on both RegDB and SYSU-MM01 datasets.

**Weaknesses:**

1. As a data augmentation strategy, it will be better if the author can provide several visualization examples.
2. In Table 4, I wonder about the detailed augmentation strategy that contains in the mentioned 'ours'. In DEEN, it already contains grayscale transformation and random erasing. So, how do you insert the RLE in such a structure?

**Questions:**

Please check the weakness part for more details.

**Limitations:**

The authors have discussed the limitations of the proposed method in Section 5.

---

> ### Author Rebuttal · Authors · 2024-08-05
>
> Thanks for your constructive and positive feedback which inspired us a lot. Below, we respond to your key concerns point by point.
>
> >**Q1: As a data augmentation strategy, it will be better if the author can provide several visualization examples.**
>
> **R1:** Thanks for your suggestion. We have provided some visualization results in Figure 1 of the attached PDF version rebuttal following your suggestion and added this part in the final-version paper. Therefore, please kindly refer to the attached PDF. Through the visualization results, we can clearly observe the differences between MRLE and RRLE. Specifically, MRLE aims to provide diverse linear augmentation results that strictly adhere to the initial reflection correlation prior, while RRLE aims to provide diverse linear augmentation results under limited risk-taking.
>
> >**Q2: In Table 4, I wonder about the detailed augmentation strategy that contains in the mentioned 'ours'. In DEEN, it already contains grayscale transformation and random erasing. So, how do you insert the RLE in such a structure?**
>
> **R2:** We feel sorry for the inconvenience caused. Since the DEEN [1] already contains the random grayscale and random erasing for data augmentation, we remove the random grayscale and add the RLE just like what we applied for the baseline method in Tbl. 1. Following your suggestion, we have strengthened this part to make it more clear.
>
> [1] Diverse embedding expansion network and low-light cross-modality benchmark for visible-infrared person re-identification. CVPR 2023.
>
> **Finally**, we hope our response has adequately addressed your concerns. If you have any further suggestions, please feel free to discuss them with us. We will provide the corresponding results as soon as possible.

---

> > ### Comment · Reviewer_DZyN · 2024-08-10
> >
> > Thanks for your response. My concerns are addressed well thus I raise my score.
> > I also looked through other reviewers' comments. And I have a different opinion from SzQ4. Though data enhancement has been explored in the ReID field for many years, there is still a lot of work in this area, especially for VI-ReID. Which enhancement or augmentation is more effective is also an important topic being rarely explored. This paper not only introduces a new data augmentation method in cross-spectral ReID, but also provides a theoretical analysis on the different enhancement methods in this field. I think this type of work is worth to be encouraged.

---

### Official Review · Reviewer_gwge · 2024-07-10

**Soundness:** 2
**Presentation:** 2
**Contribution:** 2
**Rating:** 5
**Confidence:** 4

**Summary:**

This paper explores data augmentation strategies for cross-spectral re-identification. The authors find that non-linear modal differences arise mainly from different linear transformations occurring on various material surfaces; all data enhancement strategies for cross-spectral re-identification aim to simulate such transformations. Based on these observations, they introduce a Random Linear Enhancement (RLE) augmentation method and further extend the boundaries of the moderate and radical transformations by Moderate Random Linear Enhancement (MRLE) and Radical Random Linear Enhancement (RRLE).

**Strengths:**

In this paper, by introducing the Lambertian model, the analysis finds that non-linear modal differences arise mainly from different linear transformations occurring on various material surfaces, which is an intuitive conclusion.

The RLE method proposed by the authors together with the existing random erasing achieved good experimental results.

**Weaknesses:**

Innovation is limited, and although the authors' introduction of the physical model is striking, the conclusions obtained can not actually guide the authors well in designing their methodology.

The experimental results in this paper do not reach SOTA, and the article lacks a comparison with the SOTA method [1] [2].

The ablation experiments in the article are not comprehensive, lacking experiments on smaller $t_{min}$ than 0.1 as well as experiments on single terms in mixed transformations.

The paper lacks an analysis of the performance improvement due to the joint use of random erasing.

**Questions:**

See the weakness.

**Limitations:**

There is no analysis of limitation in the paper. As mentioned in weakness, the correlation between the observations based on the physical model in this paper and the subsequent methods in the article is not strong, and the results of the article did not reach a SOTA level, and the generalisability of the data augmentation methods proposed in the paper deserves further validation.

---

> ### Author Rebuttal · Authors · 2024-08-05
>
> Thanks for your constructive feedback which helps us a lot. Below, we respond to your key concerns point by point.
>
> >**Q1: Innovation is limited, and although the authors' introduction of the physical model is striking, the conclusions obtained can not actually guide the authors well in designing their methodology.**
>
> **R1:** Thank you for your comments and your high recognition of the physical model we introduced. The core contribution of our work is that it provides a new perspective of data augmentation for cross-spectral re-identification based on the reflection model. Compared to empirical data augmentation methods that pursue visual similarity, **_this is the first work that has modeled the cross-spectral transformation to guide the design of data augmentation._** It helps us to understand where the problem lies in this task and guides us to find what is needed for this task. In fact, the proposed RLE was totally designed under the guidance of the reflection prior, which has strong directive significance.
>
> As we mentioned in Section 3, the cross-spectral modality discrepancy mainly comes from the diverse local linear transformation on different material surfaces. Therefore, to overcome the modality discrepancy caused by the cross-spectral transformation, RLE aims to mimic such diverse linear transformations, thus encouraging the network to be robust to these transformations. Specifically, MRLE aims to provide diverse linear augmentation results that strictly adhere to the initial reflection correlation prior, while RRLE aims to provide diverse linear augmentation results under limited risk-taking.
>
> The results in Table 1 and Table 4 of this paper show that the proposed RLE brings significant performance improvements, proving that the theoretical guidance of this paper withstands scrutiny. This has also been highly recognized by Reviewer muwA, who noted, "The paper is technically sound, with an elegant motivation and formulation of the proposed method."
>
> >**Q2: The experimental results in this paper do not reach SOTA, and the article lacks a comparison with the SOTA method [1] [2].**
>
> **R2:** Thank you for your comments. However, it is regrettable that you did not provide the references for you mentioned method. In fact, as a data augmentation strategy, our proposed RLE is a plug-and-play component that can be applied to any state-of-the-art method to achieve improvements. In Table 2 and Line 280\~285, we demonstrate the RLE can boost a vanilla baseline to an impressive performance. Meanwhile, in Line 294\~304 and Table 4, we embedded RLE into the latest open-sourced method DEEN [1] and achieved significant performance enhancements, reaching new SOTA results.
>
> So, if you are interested and notice any new SOTA that we have missed, please provide a reference for related works (_preferably with open-source code due to time constraints_). We are willing to provide the performance by combining the SOTAs you mentioned with RLE.
>
> [1] Zhang, Yukang, and Hanzi Wang. "Diverse embedding expansion network and low-light cross-modality benchmark for visible-infrared person re-identification." Proceedings of the IEEE/CVF conference on computer vision and pattern recognition. 2023.
>
> >**Q3: The ablation experiments in the article are not comprehensive, lacking experiments on smaller $t_{min}$ than 0.1 as well as experiments on single terms in mixed transformations.**
>
> **R3:** Following your suggestion, we have added the experiments under more $t_{min}$ settings in Table 3(c). Meanwhile, we also added experiments on each single term in the ablation.
> Herein, we show the results under SYSU-MM01 all-search mode as below (More detailed experimental results are provided in Table 3 and Table 4 of the PDF version rebuttal).
>
> $t_{min}$:
> | $t_{min}$ | R-1 | mAP | mINP |
> |:------|:-------:|------:|-----:|
> | 0.3 | 72.9 | 70.5 | 58.2 |
> | 0.2 | 73.8 | 71.3 | 59.0 |
> | **0.1** | **74.2** | **71.8** | **60.4** |
> | 0.01 | 73.8 | 71.7 | 59.8 |
> | 0.001 | 73.6 | 71.3 | 59.5 |
>
> **single terms**:
> | Method | R-1 | mAP | mINP |
> |:------|:-------:|------:|-----:|
> | R only | 65.1 | 62.8 | 49.7 |
> | G only | 65.2 | 63.0 | 50.1 |
> | B only | 63.4 | 61.5 | 48.4 |
> | **MRLE** | **70.2** | **67.0** | **53.5** |
>
> >**Q4: The paper lacks an analysis of the performance improvement due to the joint use of random erasing.**
>
> **R4:** Thank you for your comment. As explained in Lines 271-274, we have clarified why RLE can be used in conjunction with random erasing. However, we are willing to strengthen this part based on your suggestion. Specifically, although 'RE' can be considered a special case of RRLE with a linear factor of 0, RRLE encourages images to undergo more times of transformations while preventing the loss of information. Therefore, RRLE and RE have different valuation perspectives and can be effectively used together.
>
> >**Q5: There is no analysis of limitation in the paper.**
>
> **R5:** Thanks for your suggestion. In fact, we have discussed the limitation of the RLE in Sec. 6 ‘Limitations and Broader Impact’. But we are willing to strengthen related discussion following your suggestion. In general, the core limitation of RLE is that it is founded based on a Lambertain model under cross-spectral conditions. Therefore, it may limit its adaptability to other scenarios. Under extremely bad weather such as heavy rain, fog, or limited illumination, it may show limited improvement since the Lambertain model may not work well. Under these conditions, it may be necessary to combine the RLE with advanced image deraining/dehazing or illumination enhancement strategies. However, these limitations are not the main challenge we discussed in this paper but may push forward future works to explore those complex circumstances.
>
> **Finally,** we hope our response has adequately addressed your concerns. If you have any further suggestions, please feel free to discuss them with us. We will provide the corresponding results as soon as possible.

---

> > ### Comment · Reviewer_gwge · 2024-08-11
> >
> > Thank you to the authors for responding to my concerns, sorry for not stating specific sota methods in the comments, now add that it is not appropriate for you, the authors' replies addressed most of my concerns, so I will raise my score to Borderline Accept.

---

### Official Review · Reviewer_muwA · 2024-07-11

**Soundness:** 3
**Presentation:** 3
**Contribution:** 3
**Rating:** 7
**Confidence:** 5

**Summary:**

This paper presents a unified perspective that reconsiders data augmentation strategies in cross-spectral re-identification. The authors identify that the main source of cross-spectral modality discrepancies stems from various local linear transformations due to material diversity. To address this, the authors propose a novel Random Linear Enhancement (RLE) strategy, effectively leveraging this unified perspective. Experimental results demonstrate that the proposed method significantly improves the visible-infrared re-identification performance.

**Strengths:**

1.	The novel approach of considering cross-spectral data augmentation from the perspective of the reflection model is well-motivated and innovative.
	2.	The paper is technically sound, with an elegant motivation and formulation of the proposed method.
	3.	Extensive testing on various benchmark datasets, including SYSU-MM01 and RegDB, showcases the method’s effectiveness and robustness compared to state-of-the-art methods.

**Weaknesses:**

(1) Some details are simplified in the paper. For instance, visualization results of the proposed RLE are not shown, and the baseline structure could be included in the appendix.
(2) In MRLE, the sum of lambda_r, lambda_g, and lambda_b is constrained to 1, which may seem unnecessary from the basic formulation perspective. Alternative values like [0.2, 0.2, 0.5] might also satisfy the proposed perspective.
(3) This paper could do a better work by citing, comparing, and building some connections with more recently-published literatures in the ReID community such as “Robust Object Re-identification with Coupled Noisy Labels”.

**Questions:**

Please see the weaknesses :)

**Limitations:**

The authors point out the potential limitations of the proposed method.

---

> ### Author Rebuttal · Authors · 2024-08-05
>
> Thanks for your positive and constructive feedback which inspired us a lot. Below, we respond to your key concerns point by point.
>
> >**Q1:  Some details are simplified in the paper. For instance, visualization results of the proposed RLE are not shown, and the baseline structure could be included in the appendix.**
>
> **R1:**  Thanks for your suggestions. We have added some visualization results of RLE in the paper. Meanwhile, we also provide several visualization results in Figure 1 of the pdf-version rebuttal. Please kindly refer to it. Through the visualization results, we can clearly observe the differences between MRLE and RRLE. Specifically, MRLE aims to provide diverse linear augmentation results that strictly adhere to the initial reflection correlation prior, while RRLE aims to provide diverse linear augmentation results under limited risk-taking.
>
> >**Q2: In MRLE, the sum of $\lambda_r$, $\lambda_g$, and $\lambda_b$ is constrained to 1, which may seem unnecessary from the basic formulation perspective. Alternative values like [0.2, 0.2, 0.5] might also satisfy the proposed perspective.**
>
> **R2:** As shown in Fig. 3, the linear transformation brings limited modality discrepancy. So, the transformation under [0.2, 0.2, 0.5] can also be considered as a linear transformation after the transformation under [0.22, 0.22, 0.56], which will bring limited difference. Therefore, we may only need to consider the mix percentage of different channels. We also provide the related experimental results on the SYSU all-search mode below to show the influence of the sum constraint (More detailed experimental results are provided in Table 2 of the PDF version rebuttal).
>
> | Sum Constraint| R-1 | mAP | mINP |
> |:------:|:-------:|------:|-----:|
> | | 66.3 | 63.8 | 50.3 |
> | &#10004; | **70.2** | **67.0** | **63.5** |
>
> Since removing the sum constraint may lead to some bad cases (e.g. all parameters are very small), the performance is worse than the strategy with the sum constraint.
>
> >**Q3: This paper could do a better work by citing, comparing, and building some connections with more recently-published literatures in the ReID community such as “Robust Object Re-identification with Coupled Noisy Labels”.**
>
> **R3:** Thanks for your suggestion. We have included several recent related works in Sec. 2 including this paper to make our work more comprehensive. If you have any other recommended works that are related to this paper, you can provide them in the discussion. We are willing to add them in.
>
> **Finally,**  we hope our response has adequately addressed your concerns. If you have any further suggestions, please feel free to discuss them with us. We will provide the corresponding results as soon as possible.

---

> > ### Comment · Reviewer_muwA · 2024-08-12
> >
> > Thank you for your response.
> >
> > My concerns have been addressed. I believe this work could offer valuable insights to the community, particularly in highlighting the potential of focusing on the physical characteristics of the device as a promising research direction. Therefore, I would like to raise my score.

---

### Official Review · Reviewer_SzQ4 · 2024-07-12

**Soundness:** 2
**Presentation:** 3
**Contribution:** 2
**Rating:** 3
**Confidence:** 5

**Summary:**

This paper presents a novel approach to addressing the challenges in cross-spectral Re-ID, particularly the modality discrepancy between visible (VIS) and near-infrared (NIR) images. The authors propose a unified perspective based on the Lambertian reflection model to understand and categorize data augmentation strategies. This model helps in analyzing how local linear transformations can bridge the gap between different spectral images.

**Strengths:**

**Originality:** The paper introduces a novel perspective on cross-spectral Re-ID by leveraging the Lambertian reflection model. This approach provides a unique and insightful framework for understanding the modality discrepancy between visible (VIS) and near-infrared (NIR) images.

**Clarity:** The authors provide a thorough background on cross-spectral Re-ID and the Lambertian reflection model, ensuring that readers can grasp the context and significance of their contributions.

**Significance:**  By offering a unified perspective on data augmentation and introducing the RLE strategy, the authors address a critical challenge in surveillance and other applications involving spectral images.

**Weaknesses:**

(1) The paper lacks a detailed analysis of the sensitivity of the RLE strategy to its parameters involved in RLE, MRLE, and RRLE. Understanding how different parameter settings impact performance is crucial for practical deployment.

(2) While the paper demonstrates performance improvements, there is limited discussion of the real-world applicability and potential deployment challenges of RLE strategies.

(3) The paper focuses on the proposed RLE strategy but does not provide a detailed comparative analysis with other advanced data augmentation techniques beyond traditional methods.

(4) The reliance on the Lambertian reflection model may limit the generalizability of the proposed methods to scenarios where this model does not hold.

**Questions:**

(1) Why is [$\lambda_r$, $\lambda_g$, $\lambda_b$] set to [0.299, 0.587, 0.114]? The authors should perform as many ablation experiments as possible.

(2) How sensitive is the performance of the RLE strategy to different parameter settings? Have the authors explored the impact of varying these parameters systematically?

(3) How does the RLE strategy compare with other advanced data augmentation techniques in the context of cross-spectral re-identification?

(4) How applicable is the RLE strategy in scenarios where the Lambertian reflection model does not hold? Have the authors tested the method in such conditions?

(5) Can the authors provide a more detailed explanation of how the RLE strategy works?

(6) Figure 2 is very similar to PCB, can the authors do a comparative analysis accordingly?

**Limitations:**

The length of the related work is obviously insufficient. Meanwhile, the data enhancement has been explored in the Re-ID field for many years, the novelty is not very impressive.

---

> ### Author Rebuttal · Authors · 2024-08-05
>
> Thanks for your constructive feedback. Below, we respond to your key concerns point by point.
>
> >**Q1: Why is [lambda_r, lambda_g, lambda_g] set to [0.299, 0.587, 0.114]?**
>
> **R1:**  In fact, you may have misunderstood MRLE. As mentioned in Line 180~183, the [0.299, 0.587, 0.114] are not adopted for MRLE but come from the 'transforms.RandomGrayscale' function in the torchvision library. This is a classic grayscale method that previous works widely used. However, such transformation shows limited transformation results. Therefore, as described in Lines 184-194 and Eq. (6), we proposed MRLE, where the $\lambda_r$, $\lambda_g$, and $\lambda_b$ are randomly sampled from a Beta distribution in every transformation. It provides more diverse augmentation during training, enhancing the model's capability. The related experiments in Tbl. 1 demonstrate the effectiveness of the proposed MRLE. We hope this response clarifies our approach and the rationale behind it.
>
> >**Q2: How sensitive is the performance of the RLE strategy to different parameter settings? Have the authors explored the impact of varying these parameters systematically?**
>
> **R2:**  Yes, we have explored different parameters and shown results in Table 3 of the paper. The results indicate that RLE is sensitive to parameter selection. For instance, $\beta_m$ can lead to a 2.4% increase in the R-1. It is important to note that $\beta_m$ and $\beta_r$ control the shape of the sampling function, while $t_{min}$ controls the stopping time. We selected the most suitable parameters based on ablation experiments.
>
> >**Q3: How does the RLE strategy compare with other advanced data augmentation techniques?**
>
> **R3:**  Following your suggestion, we have included a comparison of our proposed method with the recently released CAJ [1] on the SYSU-MM01 dataset. The results, shown below, prove the effectiveness of our approach. We will add these experiments to the paper to make our conclusions more robust (More detailed experimental results are provided in Table 1 of the PDF rebuttal).
> | Method | R-1 | mAP | mINP |
> |:------|:-------:|------:|-----:|
> | CAJ | 73.5 | 69.4 | 55.4 |
> | **Ours** | **75.4** | **72.4** | **60.9** |
>
> >**Q4: How applicable is the RLE in scenarios where the reflection model does not hold?**
>
> **R4:** Thank you for the interesting question. Generally, the Lambertian model is suitable for most real-world applications, and major datasets, which makes RLE work well in both datasets. But the problem you mentioned is still a good suggestion. Since the Lambertian model may not work well when facing strong specular reflection or complex weather environments. Under these conditions, combining RLE with advanced image deraining/dehazing or light enhancement methods may be necessary. Your suggestion indeed points us toward future research directions, and we will include this discussion in the limitation section. Thank you again for your constructive feedback.
>
> >**Q5: Can the authors provide a more detailed explanation of how the RLE works?**
>
> **R5:** Thank you for your question. As mentioned in Sec. 3, the cross-spectral modality discrepancy mainly comes from the local linear transformation on different surfaces. Therefore, to overcome the modality discrepancy caused by this phenomenon, RLE aims to mimic such a linear transformation to provide a more diverse training set, thus encouraging the network to be robust to such a transformation. Meanwhile, the detailed processing of RRLE is also provided in Appendix A.1.
>
> >**Q6: Fig. 3 is very similar to PCB, can the authors do a comparative analysis accordingly?**
>
> **R6:**  The similarity you noted might be due to both PCB and Fig. 3 employing a horizontal segmentation strategy. However, Fig. 3 is not related to PCB. Fig. 3 presents a validation visualization experiment aimed at showing that the modality gap results from diverse local linear transformations on the image. As mentioned in Lines 160~165, we uniformly segmented 100 randomly selected images into six parts from top to bottom and multiplied each part by a linear factor. These images were then fed into ResNet-50 to observe the differences in feature space before and after the transformation. This transformation is directly applied to the image. On the other hand, PCB is a model structure that segments the features extracted from ResNet to obtain fine-grained details.
>
> >**Q7: The length of the related work is obviously insufficient.**
>
> **R7:**  Thank you for pointing out this issue. Following your suggestion, we have included additional classic and recent related works in Sec. 2 to make it more comprehensive. If you have any recommended works, please provide them in the discussion. We are willing to include them.
>
> >**Q8: The data enhancement has been explored for many years, but the novelty is not very impressive.**
>
> **R8:**  Our core contribution lies in providing a unified perspective to explain the discrepancy caused by the cross-spectral transformation, rather than purely focusing on data augmentation. **_This is the first work in this area to provide new insight that will guide us to understand which kind of data augmentation is needed and how to design strategies to overcome the cross-spectral modality discrepancy_**. It even explains why complex GAN-based models [2] may not be as efficient as basic data augmentations in this context [1]. We believe RLE is just one attempt at using this perspective to design a data augmentation strategy. It may inspire more image generation or data augmentation strategies and even network designs.
>
> [1] Channel augmentation for visible-infrared re-identification. TPAMI 2023.
>
> [2] RGB-infrared cross-modality person re-identification via joint pixel and feature alignment. ICCV 2019.
>
> **Finally**, we hope our response has adequately addressed your concerns. If you have any further suggestions, please feel free to discuss them with us. We will provide the corresponding results as soon as possible.

---

> ### Comment · Reviewer_SzQ4 · 2024-08-14
> **Reply to Authors**
>
> Thank  you  for  your  detailed  responses  to  the  raised concerns.  I  have  reviewed  your  answers and other reviewers' comments, and appreciate  the  effort  you  have  put  into  addressing  each  point.
>
> Below  are  my  thoughts  on  your  responses：
>
> Firstly, according to the author's statement, the core of the method in this paper is the addition of Beta distribution on a classic grayscale method. Meanwhile, the paper lacks sufficient theoretical demonstration, proof of derivation, and adequate experimental evidence. Therefore, I think the novelty of this paper is limited and not up to the level of NeurIPS, because NeurlPS as a top conference requires high theoretical innovation and sufficient experimental validation. I agree with Reviewer DZyN's statement "The Importance of Data Enhancement Research for VI-ReID", and as stated above, whilst the paper's methodology has significant outcomes, the depth of the research is not deep enough and the evidence for it is not sufficient.
>
> Secondly, I have looked carefully at the authors' and other reviewers' COMMENTS and I still have the following questions:
>
> (1) $\lambda_r$, $\lambda_g$, and $\lambda_b$ are randomly sampled from a Beta distribution in every transformation, thus the results are random in nature, and whether the authors conducted multiple experiments and then averaged them?
>
> (2) This paper is built around data-augmented randomness, and the author could be asked to explain specifically what this randomness benefits actually are？
>
> (3) I realized that Table 3 explored the sensitivities of $\beta_m$, $\beta_r$, and $t_{min}$, and I'm curious is the sensitivities of other hyper-parameters such as $\alpha$, $r_{min}$, $r_{max}$, $s_{min}$, $s_{max}$, etc.
>
> (4) Although the proposed method is about data enhancement, the author's comparison and analysis of the experimental results should highlight the advantages of data enhancement. There are many methods related to data enhancement, and it is obviously not enough to compare only the CAJ method, why don't the authors compare other classical and commonly used data enhancement methods?
>
> (5) This paper focuses on the Lambertian model, but this model is unable to deal with the effects of highlights and ambient light due to specular reflections on surfaces. For this problem, I think the Phong model is more appropriate, so I don't think this work has been studied enough.
>
> (6) The  distinction  between  PCB  and  your  Fig.  3  is  now  clear.  It  is  important  to  highlight  this  difference  in  the  paper  to  avoid  any  confusion  for  readers.  A  brief  comparison  or  note  in  the  figure  legend  might  be  beneficial. And, expanding  the  related  work  section  is  a  positive  step.  Please  ensure  that  the  additional  references  are  thoroughly  integrated  and  that  the  discussion  reflects  the  broader  context  of  the  field.
>
> Therefore, I think this paper leaves a lot to be desired, and I'll reduce my score to Reject.

---

> ### Author Response · Authors · 2024-08-14
>
> **Thanks for your response, but it is too late. It is only 20 minutes left for us to give a short response.**
>
> (1) For the randomness, there is no doubt that diverse augmentation is better than stable one. This has been widely verified by a large amount of data augmentation strategies such as Mixup, Random Erasing, and so on.
>
> (2) Other hyper-parameters you mentioned follow the basic setting of Random Erasing.
>
> (3) It will be better if the author can give us the specific augmentation strategy.CAJ is the most recently published data augmentation work in this area.

---

> > ### Comment · Reviewer_SzQ4 · 2024-08-14
> > **Reply to Authors**
> >
> > After reading the responses, the author doesn't fully address all of my questions, I will keep my score.

---

### Author Rebuttal · Authors · 2024-08-06

Dear Reviewers,

We would like to thank all reviewers for providing constructive feedback that helped us improve the paper. We are encouraged that reviews think our paper:

* "provides a unique and insightful framework for understanding the modality discrepancy between visible (VIS) and near-infrared (NIR) images" (Reviewer SzQ4).

* "an elegant motivation and formulation of the proposed method" (Reviewer muwA).

* "motivation of this paper seems to be valid and intriguing, and the proposed unified perspective makes sense in this topic" (Reviewer DZyN).

We have been working diligently on improving the paper on several fronts these days to address your critique. Specific feedbacks for every reviewer are provided below your review. **Herein, we have provided an additional  PDF version rebuttal including the visualization results and detailed experimental results you mentioned in your reviews.**

Best regards,

The Authors

---

### Author Response · Authors · 2024-08-10

Dear Reviewers,

Thanks again for your great efforts and constructive advice in reviewing this paper! As the discussion period progresses, we expect your feedback and thoughts on our reply. We put a significant effort into our response, with several new experiments and discussions. We really hope you'll consider our reply. We look forward to hearing from you, and we can further address unclear explanations and remaining concerns if any.

Best regards,

Authors

---

### Decision · Program_Chairs · 2024-09-25

**Decision:**

Accept (poster)

**Comment:**

The paper received one strong accept, one accept, one borderline accept, and one reject after rebuttal. While most reviewers agreed with the contribution of this work particularly for the VI-ReID domain, one reviewer carefully reviewed the paper from a more general perspective for the paper to be suited for NeurIPS. AC agrees with most points from both sides, and consider that for the particular VI-ReID domain, the paper's strengths overweigh its weaknesses, and therefore suggest acceptance of the paper. It will be of interests to researchers working on the VI-ReID domain. Authors are required to address the review comments and include the rebuttal materials to the camera-ready version. Especially, a number of Reviewer SzQ4's concerns are valid and include those discussions will definitely strengthen the current paper.